# Token Prediction as Implicit Classification to Identify LLM-Generated Text

**Yutian Chen[†], Hao Kang[†], Yiyan Zhai, Liangze Li, Rita Singh, Bhiksha Raj**

Carnegie Mellon University

Pittsburgh, PA 15213

{yutianch, haok, yiyanz, liangzel, rsingh, bhiksha}@andrew.cmu.edu

## Abstract

This paper introduces a novel approach for identifying the possible large language models (LLMs) involved in text generation. Instead of adding an additional classification layer to a base LM, we reframe the classification task as a next-token prediction task and directly fine-tune the base LM to perform it. We utilize the Text-to-Text Transfer Transformer (T5) model as the backbone for our experiments. We compared our approach to the more direct approach of utilizing hidden states for classification. Evaluation shows the exceptional performance of our method in the text classification task, highlighting its simplicity and efficiency. Furthermore, interpretability studies on the features extracted by our model reveal its ability to differentiate distinctive writing styles among various LLMs even in the absence of an explicit classifier. We also collected a dataset named `OpenLLMText`, containing approximately 340k text samples from human and LLMs, including GPT3.5, PaLM, LLaMA, and GPT2.

## 1 Introduction

In recent years, generative LLMs have gained recognition for their impressive ability to produce coherent language across different domains. Consequently, detecting machine-generated text has become increasingly vital, especially when ensuring the authenticity of information is critical, such as legal proceedings.

Traditionally, techniques like logistic regression and support vector machines (SVM) have been used for detection tasks, as explained by Jawahar et al. (2020). The analysis of textual features like perplexity is proven to be effective as well (Wu et al., 2023). Recent advancements have introduced the use of language model itself to detect generated text, such as the AI Text Classifier released by OpenAI (2023) and Solaiman et al. (2019).

However, the exponential growth in the number of parameters from hundreds of millions to hundreds of billions has significantly improved the text generation quality, presenting an unprecedented challenge to the detection task. To overcome this challenge, we propose using the inherent next-token prediction capability of the base LM for detection task, aiming not just to determine whether or not the text is generated but also to identify its source.

## 2 Related Work

### 2.1 Generated Text Detection

Learning-based approaches to machine-generated text detection can be broadly classified into two categories: unsupervised learning and supervised learning. Unsupervised learning includes GLTR developed by Gehrmann et al. (2019) that uses linguistic features like top-$k$ words to identify generated text. Another unsupervised approach, DetectGPT by Mitchell et al. (2023), employs a perturbation-based method by generating a modifications of the text via a pre-trained language model and then comparing log probabilities of original and perturbed samples. Supervised Learning includes GROVER (Zellers et al., 2020) that extracts the final hidden state and uses a linear layer for prediction in a discriminative setting. Energy-based models (Bakhtin et al., 2019) have also been investigated for discriminating between text from different sources. Solaiman et al. (2019) fine-tuned RoBERTa model on GPT-2 generated text, resulting in an accuracy of 91% on GPT2-1B in `GPT2-Output` dataset.

### 2.2 Text-to-Text Transfer Transformer

Text-to-Text Transfer Transformer (T5) (Raffel et al., 2020) has gained recognition due to its simplicity by converting all text-based language problems into a text-to-text format. Raffel et al. and Jiang et al. (2021) have shown that T5 out-performs BERT-based models on various natural language processing tasks.

However, prior approaches have not emphasized the use of T5 in the task of distinguishing the language model responsible for text generation. Furthermore, existing approaches have not directly leveraged the next-token prediction capability of

---

[†]Two authors contribute equally to this work.

[⋆]The implementation of the classification model, training process, and dataset collection is publicly available on https://github.com/MarkChenYutian/T5-Sentinel-public

the model for this particular task. Our approach advances the field by choosing T5 model as the base LM and using its next-token prediction capability to improve the accuracy and efficiency of distinguishing the origin of the text.

# 3 Dataset

## 3.1 Data Collection

The dataset we collected, named `OpenLLMText`, consists of approximately 340,000 text samples from five sources: Human, GPT3.5 (Brown et al., 2020), PaLM (Chowdhery et al., 2022), LLaMA-7B (Touvron et al., 2023), and GPT2-1B (GPT2 extra large) (Radford et al., 2019). The `OpenLLMText` dataset, along with the response collected from OpenAI & GPTZero, is publicly available on Zenodo.[1]

Human text samples are obtained from the `OpenWebText` dataset collected by Gokaslan and Cohen (2019). GPT2-1B text samples stem from `GPT2-Output` dataset released by OpenAI (2019). As for GPT3.5 and PaLM, the text samples are collected with prompt "*Rephrase the following paragraph by paragraph: [Human_Sample]*". But instructing LLaMA-7B to rephrase human text samples is ineffective, due to the lack of fine-tuning for instruction following of LLaMA-7B. Hence, we provided the first 75 tokens from the human samples as context to LLaMA-7B and obtained the text completion as the output. For further details, including the temperature and sampling method for each source, please refer to the table 3 in Appendix A.

We partitioned the `OpenLLMText` into train (76%), validation (12%) and test (12%) subsets. The detailed breakdown is listed in Table 4 in Appendix A.

## 3.2 Data Preprocessing

We noticed stylistic differences among different models. For instance, LLaMA generates \\n for newline character instead of \n as in other sources. To address the inconsistency, we followed a similar approach as Guo et al. (2023) to remove direct indicator strings and transliterate indicator characters.

## 3.3 Dataset Analysis

To avoid potential bias and shortcuts that can be learned by model unexpectedly, we analyzed

---

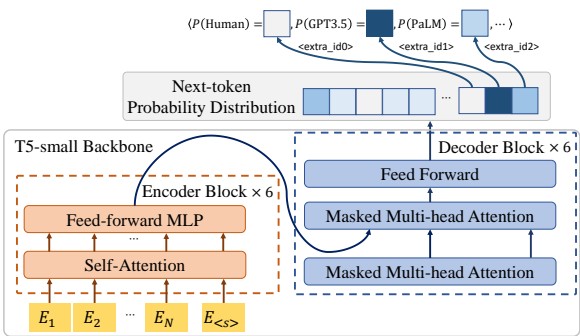

Figure 1: T5-Sentinel model architecture

the distribution of length, punctuation, tokens and word classes in `OpenLLMText` across different sources. Results indicate that no significant bias exists between different sources. For detailed analysis and visualization, please refer to the Appendix A.

# 4 Method

Our approach can be formulated as follows. Let $\Sigma$ represent the set of all tokens. The base LM can be interpreted as a function $\text{LM} : \Sigma^* \times \Sigma \to \mathbb{R}$. Given a string $s \in \Sigma^*$ and a token $\sigma \in \Sigma$, $\text{LM}(s, \sigma)$ estimates the probability of the next token being $\sigma$. Let $Y$ denote the set of labels in `OpenLLMText`, which contains "Human", "GPT-3.5", etc. We establish a bijection $f : Y \to \mathcal{Y}$, where $\mathcal{Y} \subset \Sigma$ acts as a proxy for the labels. By doing so, we reformulate the multi-class classification task $\Sigma^* \to Y$ into a next-token prediction task $\Sigma^* \to \mathcal{Y}$. Hence, the multi-class classification task can be solved directly using $\text{LM}$:

$$\hat{y} = f^{-1} \left( \arg\max_{y \in \mathcal{Y}} \text{LM}(s, y) \right) \quad (1)$$

## 4.1 T5-Sentinel

T5-Sentinel is the implementation of our approach using T5 model. Unlike previous learning-based approaches where final hidden states are extracted and passed through a separate classifier (Solaiman et al., 2019; Guo et al., 2023), T5-Sentinel directly relies on the capability of the T5 model to predict the conditional probability of next token. In other words, we train the weight and embedding of the T5 model and encode the classification problem into a sequence-to-sequence completion task as shown in Figure 1.

We use reserved tokens that do not exist in the text dataset as $\mathcal{Y}$. During fine-tuning, we use the

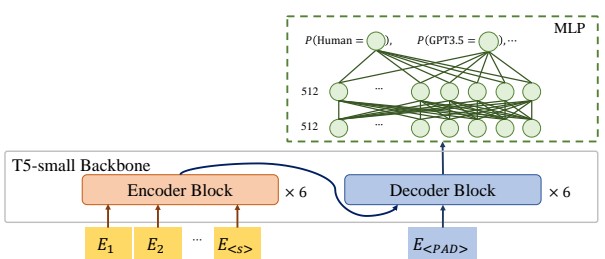

Figure 2: T5-Hidden model architecture

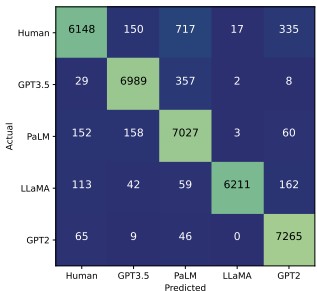

Figure 3: Confusion matrix of T5-Sentinel

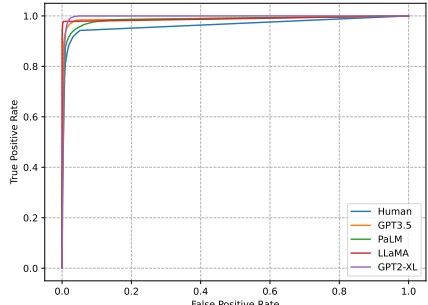

Figure 4: ROC curves for T5-Sentinel for each one-vs-rest classification task

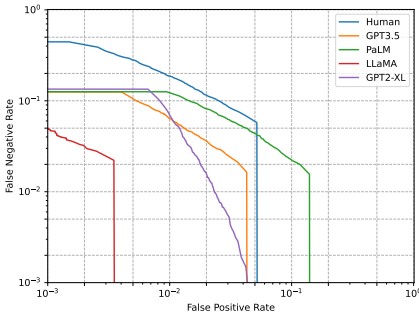

Figure 5: DET curves for T5-Sentinel for each one-vs-rest classification task

AdamW optimizer (Loshchilov and Hutter, 2017) with a mini-batch size of 128. The learning rate is $1 \times 10^{-4}$ with weight decay of $5 \times 10^{-5}$, and we train for 15 epochs.

## 4.2 T5-Hidden

To evaluate the effectiveness of not using an additional classifier while accomplishing the same classification task, we also fine-tuned the T5 model with a classifier attached, denoted *T5-Hidden*.

As illustrated with Figure 2, the classifier in T5-Hidden uses the final hidden state from the decoder block of the T5 model and computes the probability for each label after taking a softmax over its output layer. T5-Hidden is trained under identical configuration as T5-Sentinel.

## 5 Evaluation

T5-Sentinel and T5-Hidden are evaluated on the test subset of OpenLLMText dataset with receiver operating characteristic (ROC) curve, area under ROC curve (AUC) and F1 score.

### 5.1 Multi-Class Classification

We breakdown the evaluation on multi-class classification, i.e., identify the specific LLM responsible for text generation, into one-vs-rest classification for each label.

As presented in Table 5, T5-Sentinel achieves a superior weighted F1 score of 0.931 compared

to 0.833 of T5-Hidden, both under the probability threshold of 0.5. The confusion matrix for T5-Sentinel is presented in Figure 3. To illustrate the performance under different probability threshold, we plot the ROC curves and Detection Error Tradeoff (DET) curves on each one-vs-rest task in figure 4 and 5 respectively.

### 5.2 Human-LLMs Binary Classification

For generated text detection task, we compare T5-Sentinel against T5-Hidden and two widely-adopted baseline classifiers, the AI text detector by OpenAI and ZeroGPT.

Figure 6 displays the ROC curves obtained from our experiments and the detailed performance metrics such as AUC, accuracy, and F1 score are summarized in Table 1. Additionally, we compare the performance of each classifier on the generation text detection subtask for each LLM source in OpenLLMText, as shown in Table 2. Notably, T5-Sentinel outperforms the baseline across all subtasks in terms of AUC, accuracy, and F1 score.

## 6 Interpretability Study

Our interpretability studies, including a dataset ablation study and integrated gradient analysis,

|           | AUC       | Accuracy  | F1        | Recall    | Precision |
|-----------|-----------|-----------|-----------|-----------|-----------|
| OpenAI    | 0.795     | 0.434     | 0.415     | **0.985** | 0.263     |
| ZeroGPT   | 0.533     | 0.336     | 0.134     | 0.839     | 0.148     |
| T5-Hidden | 0.924     | 0.894     | 0.766     | 0.849     | 0.698     |
| T5-Sentinel | **0.965** | **0.956** | **0.886** | 0.832     | **0.946** |

Table 1: Evaluation result for T5-Sentinel and T5-Hidden on Human-LLM binary classification problem comparing to that of baselines OpenAI (2023); ZeroGPT (2023) on test susbet of `OpenLLMText` dataset.

| Task            | Human v. GPT3.5 | | | Human v. PaLM | | | Human v. LLaMA | | | Human v. GPT2 | | |
|-----------------|----------|----------|----------|----------|----------|----------|----------|----------|----------|----------|----------|----------|
| Metric          | AUC      | Acc      | F1       | AUC      | Acc      | F1       | AUC      | Acc      | F1       | AUC      | Acc      | F1       |
| OpenAI          | .761     | .569     | .694     | .829     | .659     | .743     | .676     | .573     | .709     | .901     | .768     | .809     |
| ZeroGPT         | .576     | .493     | .555     | .735     | .662     | .649     | .367     | .375     | .519     | .435     | .382     | .504     |
| Solaiman et al. | .501     | .499     | .005     | .508     | .501     | .013     | .524     | .533     | .027     | .870     | .748     | .666     |
| T5-Hidden       | **.971** | **.922** | **.916** | **.964** | **.914** | **.908** | .806     | .746     | .779     | .965     | .910     | .903     |
| *std*           | *.011*   | *.022*   | *.026*   | *.020*   | *.035*   | *.033*   | *.062*   | *.084*   | *.077*   | *.019*   | *.024*   | *.017*   |
| T5-Sentinel     | .970     | .914     | .906     | .962     | .906     | .898     | **.964** | **.903** | **.901** | .965     | **.912** | **.904** |

Table 2: Evaluation results for T5-Sentinel, T5-Hidden and baselines on each specific human-to-LLM binary classification task. For T5-Hidden model, we also tested with 5 random initializations and report the standard deviation of metrics under each task in *italic*.

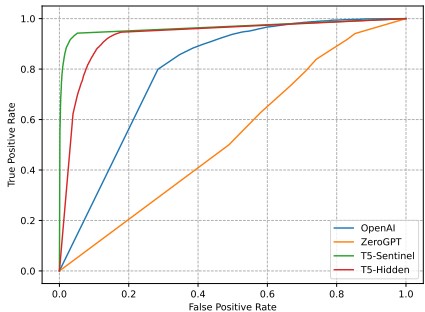

Figure 6: ROC curves for OpenAI classifier, ZeroGPT, T5-Hidden and the proposed T5-Sentinel on test subset of `OpenLLMText`

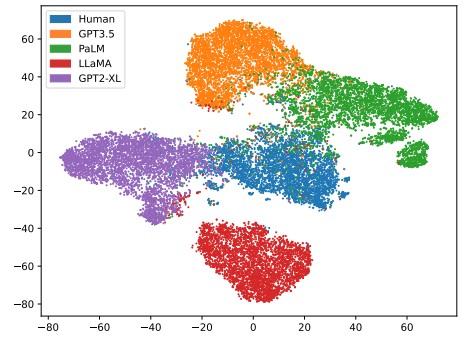

Figure 7: t-SNE plot for T5-Sentinel on test subset of `OpenLLMText` dataset under perplexity of 100

show that T5-Sentinel does not rely on unexpected shortcuts. Additionally, we employ t-distributed Stochastic Neighbor Embedding (t-SNE) projection (van der Maaten and Hinton, 2008) on the hidden states of the last decoder block of T5-Sentinel. The resulted t-SNE plot, shown in Figure 7, demonstrates the model's ability to distinguish textual contents from different sources, corroborating the evaluation results discussed earlier. For comparison, we also plotted the t-SNE plot of T5-Hidden on the test subset of `OpenLLMText` in figure 8, results show that the T5-Sentinel cannot distinguish LLaMA with other source of sample correctly. This aligns with the evaluation reported in table 2.

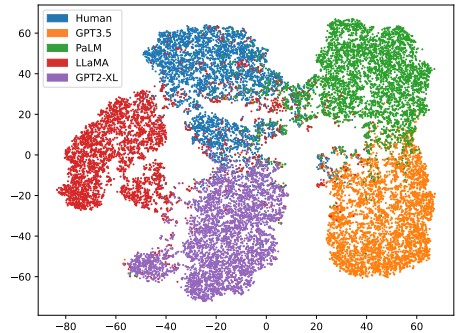

Figure 8: t-SNE plot for T5-Hidden on test subset of `OpenLLMText` dataset under perplexity of 100

## 6.1 Dataset Ablation Study

Ablation study is conducted on the `OpenLLMText` dataset to further investigate which feature is utilized by the T5-Sentinel to make classification. We design 4 different cleaning configurations on `OpenLLMText`: i) compress consecutive newline characters to one; ii) transliterate Unicode characters to ASCII characters[2]; iii) remove all punctuation; iv) cast all characters to lower case. Evaluation results for each one-vs-rest binary classification task in Table 6 shows that T5-Sentinel is quite robust to perturbations in the input text. For ablation configuration i), ii) and iv), the AUC and F1 score are almost identical. However, the performance drop significantly under condition iii) (with $\Delta\text{AUC} \approx -0.3$).

To prove that T5-Sentinel is not overfitting to specific punctuation, we independently remove each punctuation in ASCII from input text and evaluated the performance of model on each one-vs-rest classification task. Results show that only the removal of period and comma cause significant performance degradation (shown in Table 6). This can be due to the fact that T5-Sentinel is utilizing syntax structure of input sample to distinguish text from human, GPT3.5 and PaLM instead of overfitting on these two punctuation. In section 6.2, we confirm this hypothesis with an integrated gradient analysis.

## 6.2 Integrated Gradient Analysis

The integrated gradient method, proposed by Sundararajan et al. (2017), is a robust tool for attributing the prediction of a neural network to the input features. Here, we apply the integrated gradient method on the word embedding of input text sample and calculated the integrated gradient of each token using the following formula:

$$IG(x) = \frac{x - x_0}{m} \sum_{i=0}^{m} \frac{\partial L(\text{T5}(x_0 + \frac{i}{m}(x - x_0)), y)}{\partial x}$$

(2)

where $x_0$ denotes the word embedding of the input text same length as $x$ but filled with <pad> token, which is considered as a baseline input.

The visualization tool we developed uses equation 2 with $m = 100$ to calculate the integrated gradient of each token and show the attribution of each token in the prediction made by T5-Sentinel model.

Some samples for visualization can be found in appendix D.

We notice the existence of substantial gradients on non-punctuation tokens, especially on syntax structures like clauses (Sample 2, Appendix D) and semantic structures like repetitive verbs (Sample 4, Appendix D), indicating that the gradients are not exclusively overfitting on punctuation tokens. Rather, the drop in performance of the model without punctuation appears to stem from the fact that the removal of punctuation disrupts the overall semantic structure within the text that the model has correctly learned.

## 7 Conclusion

In conclusion, this paper demonstrates the effectiveness of involving next-token prediction in identifying possible LLMs that generate the text. We make contributions by collecting and releasing the `OpenLLMText` dataset, transferring the classification task into the next-token prediction task, conducting experiments with T5 model to create the T5-Sentinel, and providing insight on the differences of writing styles between LLMs through interpretability studies. In addition, we provide compelling evidence that our approach surpasses T5-Hidden and other existing detectors. As it eliminates the requirement for an explicit classifier, our approach stands out for its efficiency, simplicity, and practicality.

## Limitations

The `OpenLLMText` dataset we collected is based on the `OpenWebText` dataset. The original `OpenWebText` dataset collects human written English content from Reddit, an online discussion website mainly used in North America. Hence, the entries from human in dataset may bias towards native English speakers' wording and tone. This might lead to a degraded performance when the detector trained on `OpenLLMText` dataset is given human-written text from non-native English speakers. This tendency to misclassify non-native English writing as machine-generated is also mentioned by Liang et al. (2023).

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

## A Dataset

### A.1 Length Distribution

The length distribution of text sample from each source is presented in Figure 9. Since we truncated the text to first 512 tokens during training and evaluation, the actual length distribution for each source received by the classifier is shown in Figure 10, which is approximately the same across various sources.

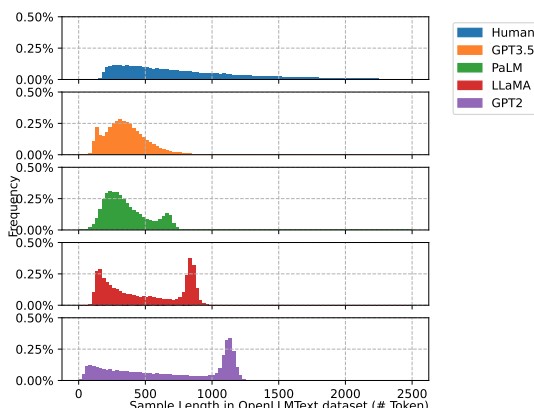

Figure 9: Distribution of sample length measured by the number of tokens in the `OpenLLMText` dataset.

## A.2 Punctuation Distribution

Figure 11 shows the distribution of top-40 ASCII punctuation in `OpenLLMText` dataset. For most of the punctuation, all LLMs tend to generate them with similar frequency. However, PaLM does tend to generate "*" more frequently than other sources. However, further experiments on dataset cleaning (indicated in 6 in Appendix C) show that T5-Sentinel is not relying on this feature to identify PaLM generated text.

## A.3 Token Distribution

The distribution of most commonly seen tokens from each source is presented in Figure 12. It is worth noting that while the GPT2 source lacks single quotation marks and double quotation marks, the overall token distributions from all sources exhibit a consistent pattern.

## A.4 Word-Class Distribution

Figure 13 displays the word-class distribution like noun, adjective and others for each source in `OpenLLMText` dataset. The distribution is almost identical across all sources.

## B Evaluation

The detailed evaluation results for human-to-LLM binary classification tasks and one-to-rest binary classification tasks are separately listed in Table 2 and 5.

## C Dataset Ablation Study

Table 6 has shown the performance of T5-Sentinel under different dataset cleaning methods.

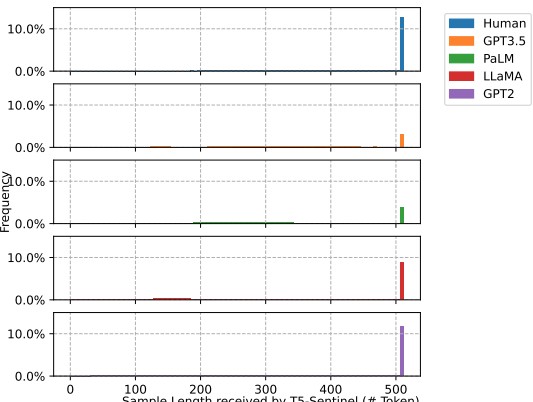

Figure 10: Distribution of sample length measured by the number of tokens actually received by T5-Sentinel during training, validation and test.

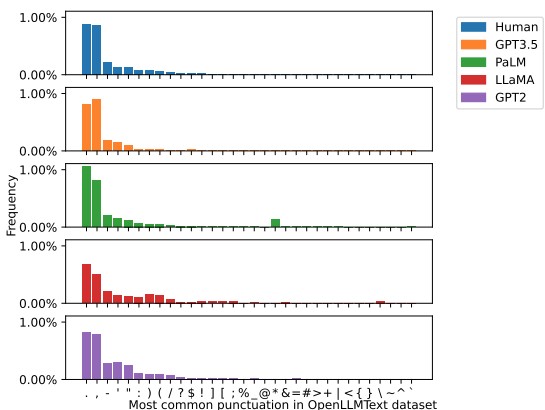

Figure 11: Distribution of ASCII Punctuations in `OpenLLMText`

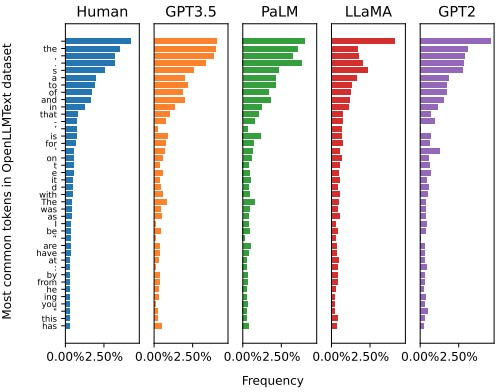

Figure 12: Distribution of tokens in `OpenLLMText`

| Source | Dataset/Tool | Generation Method | Temp | Top-$p$ |
|--------|--------------|-------------------|------|---------|
| Human | `OpenWebText` dataset | - | - | - |
| GPT3.5 | OpenAI's `gpt-3.5-turbo` API | Rephrase human samples | 1 | 1 |
| PaLM | `text-bison-001` API | Rephrase human samples | 0.4 | 0.95 |
| LLaMA | LLaMA-7B model | Text completion | 0.95 | 0.95 |
| GPT2 | `GPT2-Output` dataset | Random hidden state | 1 | 1 |

Table 3: Sources and details of text samples in `OpenLLMText`

| Subset | Human | GPT3.5 | PaLM | LLaMA-7B | GPT2-1B | Total |
|--------|-------|--------|------|----------|---------|-------|
| train | 51205 | 51360 | 46525 | 50099 | 65079 | 264268 |
| valid | 10412 | 10468 | 3485 | 9305 | 10468 | 44138 |
| test | 7367 | 7385 | 7400 | 6587 | 7385 | 36124 |
| Total | 68984 | 69213 | 57410 | 65991 | 82932 | 344530 |

Table 4: Number of entries from each source in each subset of `OpenLLMText`.

| Task | AUC | | Accuracy | | F1 | |
|------|-----|--|----------|--|----|--|
| | Sentinel | Hidden | Sentinel | Hidden | Sentinel | Hidden |
| Human v. Rest | 0.965 | 0.965 | 0.956 | 0.894 | 0.886 | 0.766 |
| GPT3.5 v. Rest | 0.989 | 0.989 | 0.979 | 0.980 | 0.949 | 0.950 |
| PaLM v. Rest | 0.984 | 0.984 | 0.957 | 0.947 | 0.901 | 0.881 |
| LLaMA-7B v. Rest | 0.989 | 0.989 | 0.989 | 0.899 | 0.969 | 0.616 |
| GPT2-1B v. Rest | 0.995 | 0.995 | 0.981 | 0.969 | 0.955 | 0.929 |
| Average (weighted) | 0.984 | 0.984 | 0.972 | 0.939 | 0.931 | 0.833 |

Table 5: Evaluation result for each one-vs-rest classification task for T5-Sentinel and T5-Hidden on `OpenLLMText` test subset. Accuracy, and F1-score are calculated under probability threshold of 0.5.

| One-vs-rest | Human | | GPT3.5 | | PaLM | | LLaMA | | GPT2 | |
|-------------|-------|--|--------|--|------|--|-------|--|------|--|
| Metric | AUC | F1 | AUC | F1 | AUC | F1 | AUC | F1 | AUC | F1 |
| **Original** | .965 | .886 | .989 | .949 | .984 | .901 | .989 | .969 | .995 | .955 |
| Newline | .965 | .886 | .989 | .949 | .984 | .901 | .989 | .969 | .995 | .955 |
| ↓ Unicode | .947 | .832 | .987 | .941 | .983 | .895 | *.981* | *.946* | .988 | .907 |
| ↓ Punc | **.775** | **.493** | **.590** | **.096** | **.679** | **.120** | **.974** | **.880** | **.942** | **.729** |
| ↓ . | *.918* | *.661* | .877 | .645 | .886 | *.619* | .993 | .954 | *.986* | .882 |
| ↓ , | .946 | .784 | .954 | .861 | .931 | .794 | .993 | .974 | .991 | .922 |
| ? | .967 | .890 | .989 | .949 | .984 | .904 | .989 | .968 | .995 | .955 |
| ! | .966 | .888 | .989 | .949 | .984 | .903 | .988 | .968 | .995 | .954 |
| : | .966 | .889 | .989 | .949 | .984 | .905 | .988 | .965 | .995 | .952 |
| ' | .969 | .884 | .988 | .948 | .983 | .903 | .989 | .968 | .994 | .946 |
| " | .966 | .881 | .988 | .947 | .983 | .901 | .985 | .961 | .995 | .951 |
| * | .964 | .881 | .988 | .946 | .978 | .891 | .989 | .968 | .995 | .953 |
| Lower | .966 | .863 | .984 | .928 | .973 | .889 | .987 | .962 | .989 | .914 |

Table 6: Evaluation results of T5-Sentinel on `OpenLLMText` dataset under different ablation configurations. "Newline", "Unicode", "Lower" and "Punc" stands for the cleaning configuration i) to iv) respectively. Each nested row under "Punc" represents removing that specific punctuation. ↓ means the accuracy drop by a considerable amount. **Bold** and *italic* represents the worst and second-worst entries in that column.

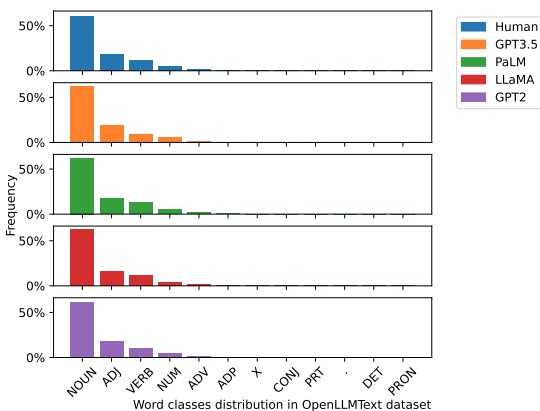

Figure 13: Distribution of word classes in OpenLLMText

## D Integrated Graident Samples

Some samples of integrated gradient results are presented below. Brighter background indicates a higher integrated gradient value on the token and meaning that specific token contributes more on final prediction result. Sample 1 - 5 are randomly chosen from the test set of OpenLLMText dataset.

---

Sample 1. Label: Human, Predicted: Human

Out going US President Barack Obama said that he did not expect President - e lect Donald Trump to follow his administration ' s blueprint s in dealing with Russia , yet hoped that Trump would " stand up " to Moscow . " My hope is that the president - e lect coming in takes a similarly constructive approach , . . . However , he repeated allegations that Russia had engaged in cyber attack s against the US . Although US intelligence officials blame d Russia for cyber attack s on the Democratic National Committee , they have not provided any substantial proof to the

(. . . Truncated)

---

Sample 2. Label: GPT3.5, Predicted: GPT3.5

Barack Obama has stated that he hopes Presi- dent - e lect Donald Trump will confront Russia , despite not expecting him to follow the cur- rent administration ' s policies . . . . W hilst he wished Russia well and acknowledged it as an important partner to the world , Obama ex- pressed hope for Trump ' s success " not just by its own people but by people around the world " . Obama commented that not everything that had worked for Trump

(. . . Truncated)

---

Sample 3. Label: PaLM, Predicted: PaLM

HTC has had a tough year . Revenue is down , it lost a patent suit to Apple , and it d re w criticism for pulling the plug on Je lly Bean updates for some of its phones . The company needs a win , and it ' s hoping that the D roid DNA will be it . The DNA is a high - end phone with a 5- inch 10 80 p display , the same quad - core Snapdragon chip as the Opti mus G and Nex us 4, and 2 GB of RAM . It ' s a powerful phone with a beautiful screen , but there are some trade off s . The first trade off is battery life . The DNA ' s battery is smaller than the batteries in the Opti mus G and Galaxy S III , and it doesn ' t last as long .

(. . . Truncated)

---

Sample 4. Label: LLaMA, Predicted: LLaMA

. . . Barack Obama s aid he did not expect P resident - e lect Donald Trump to follow his administration ' s s blueprint s in dealing with Russia , yet hoped that Tru mp would " stand up " to Moscow . speaking during a joint press ap- pearance in White House after meeting Trump . The president also said that while Americans are concerned about Russian interference in to last year ' s election campaign and what it might mean for the f u ture of democracy , there was no evidence that votes were rig ged by outside actors at any point . said Obama during a joint press appearance in White House after meeting Trump . Out going US President Barack O b a mas aid that while Americans are concerned about Russian interference in to last year

(... Truncated)

---

Sample 5. Label: GPT2, Predicted: GPT2

Gene ric and other online electric curb side me- ter sizes There have been 11 50 comparison s between generic sets of in line -3 - gau ge , sin- gle - phase and Can ten n a - type electric curb side meters in Ontario over the past 20 years (19 75 - 28 ), total ling 2 2.3 M km . Here are samples of current (10 - year average ) home electric curb side meters from selected suppli- ers . All currently available meters have a 1 " restriction and are marked with the size in deci mal format and a code of 1 or PF

(. . . Truncated)