# OpenReview forum: "Token Prediction as Implicit Classification to Identify LLM-Generated Text"
_EMNLP/2023/Conference — EMNLP 2023 Main_

### Official Review · Reviewer_GnKq · 2023-08-02

**Soundness:** 2

**Excitement:**

3: Ambivalent: It has merits (e.g., it reports state-of-the-art results, the idea is nice), but there are key weaknesses (e.g., it describes incremental work), and it can significantly benefit from another round of revision. However, I won't object to accepting it if my co-reviewers champion it.

**Paper Topic And Main Contributions:**

This paper discusses a recent hot topic of identifying whether or not a generative language model was used to generate a given text.
The proposed method reduces the problem into one of predicting the next token given a target text based on another generative model (The authors use T5 for this), which enables not only binary classification but also prediction of which of the candidate models was used.
Experimental results show that the proposed method significantly outperforms two baseline classifiers (OpenAI and ZeroGPT) and a straightforward approach of adding a classification layer on top of T5 model, in terms of F1, Accuracy, and AUC.

**Questions For The Authors:**

1. Do you have any plan to publish the OpenLLMText dataset?

2. You concluded that T5-Sentinel outperforms T5-Hidden according to the result shown in Table 1. However, T5-Hidden is better in recall. I found that you used the same threshold of 0.5 for probability (mentioned in section 5.1) for both T5-Sentinel and T5-Hidden. Maybe you should try different values of the threshold for each of T5-Sentinel and T5-Hidden to identify the best threshold for each, which would give a more precise comparison result because the optimal value of the threshold may differ between these two. Could you share any information on this if you have?

3. Many people will be interested in the t-SNE plot for T5-"Hidden" to understand the difference between T5-Sentinel and it. Could you show any result?

4. In "3.1 Data Collection," the text samples of GPT2-1B may be different in some way from the other four sets of samples (Human, GPT3.5, PaLM, and LLaMA-7B) because the other four datasets are based on the same Human text samples. Could you let me know if you have any insights on this?

**Reasons To Accept:**

The research topic is hot recently and would attract many audiences.
The dataset the authors created would potentially accelerate research and development in this field.

**Reasons To Reject:**

There are a very important but questionable point for which the authors do not have a plausible explanation. (Please see my 2nd question in Questions For The Authors)
Since the authors do not state whether or not they will make the dataset publicly available, no replication study is possible though the experimental setting is clear.

**Reproducibility:**

3: Could reproduce the results with some difficulty. The settings of parameters are underspecified or subjectively determined; the training/evaluation data are not widely available.

**Reviewer Confidence:**

3: Pretty sure, but there's a chance I missed something. Although I have a good feel for this area in general, I did not carefully check the paper's details, e.g., the math, experimental design, or novelty.

**Typos Grammar Style And Presentation Improvements:**

Throughout the paper, many contents in Appendix are mentioned as if they were in the main body of the paper. A few examples: "As illustrated with Figure 9" in line 171 (Page 3), "As presented in Table 4" in line 187 (Page 3). I would say that this is not in a polite manner, and I would recommend writing like "... Please refer to Figure 9 in Appendix for details," etc.

---

> ### Author Rebuttal · Authors · 2023-08-29
>
> 1. Do you have any plan to publish the OpenLLMText dataset?
> - After a thorough examination of the legal agreements governing ChatGPT, PaLM, LLaMA, and GPT-2, we are confident that there are no legal impediments to the public release of the constructed dataset, as long as researchers don’t use the model’s output to develop a competing product with the existing chat bot.
> - This restriction however, doesn’t undermine the reproducibility of our work, as we’re using the dataset to conduct research on identifying LLM-generated text. The dataset is presently available for anonymous access via Zenodo: https://zenodo.org/record/8285326.
> - We’re also aware that third-party classifiers mentioned in our paper like the one developed by OpenAI has removed its pubclic access, so we’ve included the raw responses from each third-party classification API in the Zenodo link as well.
> - For complete transparency, we have made the codebase accessible in an anonymous repository, encompassing model definitions, training procedures, and evaluation code. You can find it via **https://anonymous.4open.science/r/Token-Prediction-As-Implicit-Classification-For-LLM/README.md**.
> - After the anonymous review period, we plan to release the model weights of T5-Sentinel and T5-Hidden.
> ---
> 2. You concluded that T5-Sentinel outperforms T5-Hidden according to the result shown in Table 1. However, T5-Hidden is better in recall. I found that you used the same threshold of 0.5 for probability (mentioned in section 5.1) for both T5-Sentinel and T5-Hidden. Maybe you should try different values of the threshold for each of T5-Sentinel and T5-Hidden to identify the best threshold for each, which would give a more precise comparison result because the optimal value of the threshold may differ between these two. Could you share any information on this if you have?
> - We’ve already tried out different threshold ranging from 0 to 1 with step size of 0.1 on the prediction of both T5-Hidden and T5-Sentinel. Please refer to Figure 10 in Appendix B to see the resulting true positive rate and false positive rates under each threshold value are plotted as the ROC curve. And it is clear that T5-Sentinel out-performs T5-Hidden under all threshold value in terms of ROC curve.
> ---
> 3. Many people will be interested in the t-SNE plot for T5-"Hidden" to understand the difference between T5-Sentinel and it. Could you show any result?
> - We have plotted the t-SNE for T5-Hidden (with perplexity=100, same as the t-SNE plot for T5-Sentinel in the paper). Results show that T5-Hidden can clearly classify the text from Human, PaLM, GPT3.5 and GPT2. However, the hidden state vector of LLaMA from T5-Hidden is not mapped to split up with other classes correctly. This plot aligns well with the result reported in Table 5, appendix B.
> - We’ll also add this plot (https://markdown-img-1304853431.file.myqcloud.com/Screenshot%202023-08-29%20at%2019.04.25.png) to the appendix in the camera-ready version of our paper.
> ---
> 4. In "3.1 Data Collection," the text samples of GPT2-1B may be different in some way from the other four sets of samples (Human, GPT3.5, PaLM, and LLaMA-7B) because the other four datasets are based on the same Human text samples. Could you let me know if you have any insights on this?
> - Yes, this is a potential problem in the data collection process. While using rephrasing / continuation prompt to construct the dataset for generated content can make our detector focus more on the syntactic and wording difference between different sources instead of the semantic difference between samples, we found that it is hard to instruct the GPT2-XL to follow the prompt and provide high-quality generation result based on human sample provided. Therefore, we used the open-source dataset released by OpenAI where content are generated from randomly initialized hidden state vector instead.
> - Since the task we are studying is to detect if a provided sample is generated or not and the source of sample, we believe the GPT2-output dataset still fits the definition of task here.

---

### Official Review · Reviewer_K9Y3 · 2023-08-03

**Typos Grammar Style And Presentation Improvements:** 1. Section 5.1 may be put in Appendix…
**Soundness:** 2

**Excitement:**

3: Ambivalent: It has merits (e.g., it reports state-of-the-art results, the idea is nice), but there are key weaknesses (e.g., it describes incremental work), and it can significantly benefit from another round of revision. However, I won't object to accepting it if my co-reviewers champion it.

**Paper Topic And Main Contributions:**

This paper addresses a text identification task to identify texts generated by large language models (LLMs). The paper proposes a T5-based method that uses a next-token prediction head to solve the task. To evaluate the proposed method, the paper also constructs a dataset called OpenLLMText that consists of texts created by humans and 4 LLMs (GPT3.5, PaLM, LLaMA, and GPT2), totaling ~340K text samples. Experimental results demonstrate the effectiveness of the proposed method over the standard classifier-based method as well as two third-party APIs (OpenAI and ZeroGPT) and provide interpretability analyses of the proposed method.

Updated: I have read the author rebuttal and keep the scores unchanged. While it is very useful to make the datasets, code, and models available, the effectiveness of the proposed method (T5-Sentinel) is limited since there is no statistical difference between T5-Sentinel and its strong baseline (T5-Hidden) on three out of four binary classification tasks (see [the discussion](https://openreview.net/forum?id=B3SjWgXHzM&noteId=qTlHpAtqwr) with the table along with standard deviations).

**Questions For The Authors:**

A. Will the constructed dataset and predictions by models tested in the paper become publicly available?

B. Which size of T5 is used in terms of the number of parameters? Do the different sizes of T5 affect the tendency that the proposed method outperforms other baselines?

C. Similarly, how is the proposed method compared to the previous work, such as Bakhtin et al. (2019) and Solaiman et al. (2019)?

D. How can the comparison method, T5-Hidden, be visualized by the t-SNE plot compared to the proposed method? Does the comparison method also indicate the ability to distinguish textual content from different sources by the plot?

**Reasons To Accept:**

1. The proposed method uses a next-token prediction from T5, which can leverage the prior knowledge obtained during pre-training.
2. The paper compares the proposed method with not only the standard classifier-based method of T5 but also third-party methods.

**Reasons To Reject:**

1. The paper, especially the result section, is not self-contained; multiple main results that verify the main claim of the paper, such as Figures 10-12 and Table 6, are placed in Appendix.
2. It is unclear whether the constructed dataset and predictions by models tested in the paper will become publicly available.
    - Given that [OpenAI’s text classification became unavailable as of July 20, 2023](https://openai.com/blog/new-ai-classifier-for-indicating-ai-written-text), it is difficult for other researchers to reproduce the experiments. Availability of these data is encouraged and valuable for making the experiment reproducible and advancing the research on generated text identification.

**Reproducibility:**

2: Would be hard pressed to reproduce the results. The contribution depends on data that are simply not available outside the author's institution or consortium; not enough details are provided.

**Reviewer Confidence:**

4: Quite sure. I tried to check the important points carefully. It's unlikely, though conceivable, that I missed something that should affect my ratings.

---

> ### Author Rebuttal · Authors · 2023-08-29
>
> A. Will the constructed dataset and predictions by models tested in the paper become publicly available?
> - After a thorough examination of the legal agreements governing ChatGPT, PaLM, LLaMA, and GPT-2, we are confident that there are no legal impediments to the public release of the constructed dataset, as long as researchers don’t use the model’s output to develop a competing product with the existing chat bot.
> - This restriction however, doesn’t undermine the reproducibility of our work, as we’re using the dataset to conduct research on identifying LLM-generated text. The dataset is presently available for anonymous access via Zenodo: https://zenodo.org/record/8285326.
> - We’re also aware that third-party classifiers mentioned in our paper like the one developed by OpenAI has removed its pubclic access, so we’ve included the raw responses from each third-party classification API in the Zenodo link as well.
> - For complete transparency, we have made the codebase accessible in an anonymous repository, encompassing model definitions, training procedures, and evaluation code. You can find it via **https://anonymous.4open.science/r/Token-Prediction-As-Implicit-Classification-For-LLM/README.md**.
> - After the anonymous review period, we plan to release the model weights of T5-Sentinel and T5-Hidden.
> ---
> B. Which size of T5 is used in terms of the number of parameters? Do the different sizes of T5 affect the tendency that the proposed method outperforms other baselines?
> - We used the smallest version of T5 model, T5-small as the backbone of our experiment, which about contains 60M parameters.
> - We did not try other versions of the T5 model. Our rationale stems from the conviction that employing the most compact T5 version suffices to showcase the intuitive and yet efficacious nature of token prediction in discerning LLM-generated text. Furthermore, the training dataset we’ve collected and the computing resources available to us may not fully support fine-tuning on larger models. Even the second smallest version of T5 model contains about 220M parameters, which is almost 4 times larger.
> ---
> C. Similarly, how is the proposed method compared to the previous work, such as Bakhtin et al. (2019) and Solaiman et al. (2019)?
> - We managed to reproduce the work of Solaiman et al. (2019) and ran evaluation on the OpenLLMText dataset. Results show that the detector proposed by Solaiman et al. performs well only in detecting the content generated by GPT2. Yet, the model we proposed, T5-Sentinel, still outperforms the detector by Solaiman et al. by a large margin on this task.
> - We suspect the performance degradation in other tasks are due to the difference in how text is generated. In task Human v. GPT3.5, Human v. PaLM and Human v. LLaMA, the generated text are rephrased / continued from provided human written sample, while in GPT2 output dataset, the samples are randomly generated from random hidden state vector.
>
> | Task            | Human v. GPT3.5 |      |      | Human v. PaLM |      |      | Human v. LLaMA |      |      | Human v. GPT2 |      |      |
> |-----------------|-----------------|------|------|---------------|------|------|----------------|------|------|---------------|------|------|
> | Metric          | AUC             | Acc  | F1   | AUC           | Acc  | F1   | AUC            | Acc  | F1   | AUC           | Acc  | F1   |
> | OpenAI          | .761            | .569 | .694 | .829          | .659 | .743 | .676           | .573 | .709 | .901          | .768 | .809 |
> | ZeroGPT         | .576            | .493 | .555 | .735          | .662 | .649 | .367           | .375 | .519 | .435          | .382 | .504 |
> | Solaiman et al. | .501            | .499 | .005 | .508          | .501 | .013 | .524           | .533 | .027 | .870          | .748 | .666 |
> | T5-Hidden       | .971            | .922 | .916 | .964          | .914 | .908 | .806           | .746 | .779 | .965          | .910 | .903 |
> | T5-Sentinel     | .970            | .914 | .906 | .962          | .906 | .898 | .964           | .903 | .901 | .965          | .912 | .904 |
> ---
> D. How can the comparison method, T5-Hidden, be visualized by the t-SNE plot compared to the proposed method? Does the comparison method also indicate the ability to distinguish textual content from different sources by the plot?
> - We have plotted the t-SNE for T5-Hidden (with perplexity=100, same as the t-SNE plot for T5-Sentinel in the paper). Results show that T5-Hidden can clearly classify the text from Human, PaLM, GPT3.5 and GPT2. However, the hidden state vector of LLaMA from T5-Hidden is not mapped to split up with other classes correctly. This plot aligns well with the result reported in Table 5, appendix B.
> - We’ll also add this plot (https://markdown-img-1304853431.file.myqcloud.com/Screenshot%202023-08-29%20at%2019.04.25.png) to the appendix in the camera-ready version of our paper.

---

### Official Review · Reviewer_PtZE · 2023-08-05

**Typos Grammar Style And Presentation Improvements:** 1. Please consider moving some tables…
**Soundness:** 3

**Excitement:**

4: Strong: This paper deepens the understanding of some phenomenon or lowers the barriers to an existing research direction.

**Paper Topic And Main Contributions:**

The paper focuses on classifying machine-generated text. To this end, the authors proposed T5-Sentinel together with T5-Hidden detectors to distinguish human-generated text and several other LLM-generated texts. The experimental result shows that the proposed T5-Sentinel classifier outperforms T5-Hidden and other baselines. Furthermore, the dataset OpenLLMText containing 340k text samples was collected for the experimental purpose.

**Questions For The Authors:**

1. How did you determine the training/dev/test split as 76%, 12%, and 12%?
2. Have you run experiments under different random seeds to observe the performance variance?
3. Is it possible to release your proposed classifier and constructed dataset?

**Reasons To Accept:**

1. The proposed T5-Sentinel shows superior performance in distinguishing human-generated and machine-generated text compared with other baseline methods.

2. A dataset with 340k text samples is constructed, which may benefit future research in this field.

**Reasons To Reject:**

1. The statistical significance is not reported in Table 1. For example, T5-Hidden contains an additional classifier head, and how different random initialization (over different random seeds) of the additional classifier would affect the performance is unclear.

2. As one contribution, the release of the dataset is subject to the data and other LLM models' licenses. The failure of data release may diminish the value of this work.

**Reproducibility:**

3: Could reproduce the results with some difficulty. The settings of parameters are underspecified or subjectively determined; the training/evaluation data are not widely available.

**Reviewer Confidence:**

4: Quite sure. I tried to check the important points carefully. It's unlikely, though conceivable, that I missed something that should affect my ratings.

---

> ### Author Rebuttal · Authors · 2023-08-29
>
> 1. How did you determine the training/dev/test split as 76%, 12%, and 12%?
> - A common approach to training/dev/test split is 80%, 10% and 10%. However, since we’re using the smallest version of T5 model (i.e. about 60M parameters), we reduce the training ratio by a little bit.
> ---
>
> 2. Have you run experiments under different random seeds to observe the performance variance?
> - In the case of T5-Sentinel, because we’re using the pre-trained model as a direct initialization value, different random seeds should not influence the weight initialization in the training process. So we did not try different random seeds on this one.
> - For T5-Hidden, an additional classifier head in the form of an MLP (Multi Layer Perceptron) is present. This MLP undergoes initialization through the Kaiming method (Kaiming et al., 2015), where randomness is involved. Our assessment involved testing T5-Hidden’s performance across five distinct random seeds and subsequently computing the evaluation metrics detailed in Table 1.
> - In the table attached below, the value of standard deviation for evaluation metrics is presented in the parentheses.
>
> |                 | AUC           | Accuracy      | F1            | Recall        | Precision     |
> |-----------------|---------------|---------------|---------------|---------------|---------------|
> | OpenAI          | 0.795         | 0.434         | 0.415         | 0.985         | 0.263         |
> | ZeroGPT         | 0.533         | 0.336         | 0.134         | 0.839         | 0.148         |
> | T5-Hidden       | 0.924 (0.024) | 0.894 (0.010) | 0.766 (0.052) | 0.849 (0.061) | 0.698 (0.079) |
> | T5-Sentinel     | 0.965         | 0.956         | 0.886         | 0.832         | 0.946         |
> ---
>
> 3. Is it possible to release your proposed classifier and constructed dataset?
> - After a thorough examination of the legal agreements governing ChatGPT, PaLM, LLaMA, and GPT-2, we are confident that there are no legal impediments to the public release of the constructed dataset, as long as researchers don’t use the model’s output to develop a competing product with the existing chat bot.
> - This restriction however, doesn’t undermine the reproducibility of our work, as we’re using the dataset to conduct research on identifying LLM-generated text. The dataset is presently available for anonymous access via Zenodo: https://zenodo.org/record/8285326.
> - We’re also aware that third-party classifiers mentioned in our paper like the one developed by OpenAI has removed its pubclic access, so we’ve included the raw responses from each third-party classification API in the Zenodo link as well.
> - For complete transparency, we have made the codebase accessible in an anonymous repository, encompassing model definitions, training procedures, and evaluation code. You can find it via https://anonymous.4open.science/r/Token-Prediction-As-Implicit-Classification-For-LLM/README.md.
> - After the anonymous review period, we plan to release the model weights of T5-Sentinel and T5-Hidden.

---

### Meta-Review · Area_Chair_XVyr · 2023-09-17

**Recommendation:** 3

**Metareview:**

The paper presents the T5-Sentinel method for distinguishing human-generated and machine-generated text and introduces a dataset for this purpose. Reviewers agree that T5-Sentinel demonstrates superior performance in distinguishing human-generated and machine-generated text compared to baseline methods. Additionally, the construction of a dataset with 340k text samples is seen as a valuable contribution to the field. This dataset is expected to benefit future research in the area, making it another positive aspect of the paper. The relevance of the research topic to current trends is acknowledged, and it is noted that the paper is likely to attract a wide audience.

Reviewers raise concerns about the lack of reporting statistical significance in Table 1 but the authors addressed this concern during the rebuttal period. They note that certain aspects of the method, such as T5-Hidden containing an additional classifier head, could benefit from a discussion of how different random initializations might affect performance.

The potential issue of data and model licenses affecting the release of the dataset is raised as the main concern that especially impacts reproducibility. To my understanding,  there are no legal drawbacks to the public release of the constructed dataset and the model weights and the authors mention they will release them after the anonymity period. Another concern is about the paper's self-containment, particularly in the results section. Reviewers note that important results verifying the main claim of the paper are placed in the Appendix, making it less accessible to readers. The paper should ensure that key results are presented in a more prominent manner.

In summary, the paper has several positive aspects, including superior performance and dataset contribution. However, it needs to address concerns related to statistical significance, self-containment, and replication guarantees. Ensuring the clarity of data release plans and addressing these concerns could strengthen the paper's overall contribution and impact.

---

### Decision · Program_Chairs · 2023-10-07

**Decision:**

Accept-Main

**Comment:**

The paper presents the T5-Sentinel method for distinguishing human-generated and machine-generated text and introduces a dataset for this purpose. Reviewers agree that T5-Sentinel demonstrates superior performance in distinguishing human-generated and machine-generated text compared to baseline methods. Additionally, the construction of a dataset with 340k text samples is seen as a valuable contribution to the field. This dataset is expected to benefit future research in the area, making it another positive aspect of the paper. The relevance of the research topic to current trends is acknowledged, and it is noted that the paper is likely to attract a wide audience.

Reviewers raise concerns about the lack of reporting statistical significance in Table 1 but the authors addressed this concern during the rebuttal period. They note that certain aspects of the method, such as T5-Hidden containing an additional classifier head, could benefit from a discussion of how different random initializations might affect performance.

The potential issue of data and model licenses affecting the release of the dataset is raised as the main concern that especially impacts reproducibility. To my understanding,  there are no legal drawbacks to the public release of the constructed dataset and the model weights and the authors mention they will release them after the anonymity period. Another concern is about the paper's self-containment, particularly in the results section. Reviewers note that important results verifying the main claim of the paper are placed in the Appendix, making it less accessible to readers. The paper should ensure that key results are presented in a more prominent manner.

In summary, the paper has several positive aspects, including superior performance and dataset contribution. However, it needs to address concerns related to statistical significance, self-containment, and replication guarantees. Ensuring the clarity of data release plans and addressing these concerns could strengthen the paper's overall contribution and impact.